

# Urine manganese, cadmium, lead, arsenic, and selenium among autism spectrum disorder children in Kuala Lumpur

Muhammad Ridzwan Rafi'i[1], Mohd Hasni Ja'afar[1], Mohd Shahrol Abd Wahil[2] and Shahrul Azhar Md Hanif[1]

[1] Department of Public Health Medicine, Universiti Kebangsaan Malaysia, Cheras, Kuala Lumpur, Malaysia
[2] Disease Control Division, Ministry of Health Malaysia, Putrajaya, Wilayah Persekutuan Putrajaya, Malaysia

## ABSTRACT

**Background**. The development of autism spectrum disorder (ASD) may stem from exposure to environmental pollutants such as heavy metals. The primary objective of this study is to determine the role of heavy metals of concern such as manganese (Mn), cadmium (Cd), lead (Pb), arsenic (As), and essential trace element selenium (Se) among ASD children in Kuala Lumpur, Malaysia.

**Method**. A total of 155 preschoolers in Kuala Lumpur between the ages 3 to 6 participated in an unmatched case-control study, comprising ASD children ($n = 81$) recruited from an early intervention program for autism, and 74 children without autism who were recruited from public preschools. Urine samples were collected at home, delivered to the study site, and transported to the environmental lab within 24 hours. Inductively coupled plasma mass spectrometry (ICP-MS) was applied to measure the concentration of heavy metals in the samples. Data were analysed using bivariate statistical tests (Chi-square and T-test) and logistic regression models.

**Result**. This study demonstrated that Cd, Pb, and As urine levels were significantly greater in children without autism relative to those affected with ASD ($p < 0.05$). No significant difference was in the levels of Se ($p = 0.659$) and Mn ($p = 0.875$) between children with ASD and the control group. The majority of children in both groups have urine As, Pb, and Cd values lower than 15.1 µg/dL, 1.0 µg/dL, and 1.0 µg/dL, respectively which are the minimal risk values for noncarcinogenic detrimental human health effect due to the heavy metal's exposure . Factors associated with having an ASD child included being a firstborn, male, and higher parental education levels (adjusted odds ratios (aOR) > 1, $p < 0.05$).

**Conclusion**. Preschoolers in this study demonstrated low levels of heavy metals in their urine samples, which was relatively lower in ASD children compared to the healthy matched controls. These findings may arise from the diminished capacity to excrete heavy metals, especially among ASD children, thereby causing further accumulation of heavy metals in the body. These findings, including the factors associated with having an ASD child, may be considered by healthcare professionals involved in child development care, for early ASD detection. Further assessment of heavy metals among ASD children in the country and interventional studies to develop effective methods of addressing exposure to heavy metals will be beneficial for future reference.

Corresponding author
Muhammad Ridzwan Rafi'i,
p137727@siswa.ukm.edu.my

# INTRODUCTION

Autism spectrum disorder (ASD) is a neurodevelopmental disorder that affects communication, social interaction, and behaviour with a prevalence of 10 in 1,000 children worldwide (*Zeidan et al., 2022*). In Malaysia, the prevalence of ASD was estimated at 1.6 in 1,000 children almost two decades ago (*Malaysia, 2014*).

ASD is diagnosed based on the presence of behavioural and developmental symptoms such as difficulty with social interaction and communication, repetitive behaviour, and a limited range of interests or activities (*Li et al., 2022*). These symptoms typically appear in early childhood and arise from a combination of existing genetic factors and environmental factors that affect ongoing and rapid brain development (*Li et al., 2022*; *Zeidan et al., 2022*). In addition, environmental exposure to heavy metals such as lead (Pb), arsenic (As), and cadmium (Cd) is among the important risk factors contributing to the development of ASD. Environmental factors encompassing the industrial revolution, rapid urbanization, and fast economic expansion significantly impact normal body physiology, which is believed to increase the risk of ASD (*Rossignol, Genuis & Frye, 2014*). Accumulated findings from the literature suggest that heavy metals participate in the pathogenesis of ASD *via* epigenetic pathways, and exposure to these metals during childhood could trigger epigenetic effects on DNA methylation (*Yasuda, Yasuda & Tsutsui, 2013*; *Schneider, Kidd & Anderson, 2013*). Furthermore, oxidative stress and inflammatory response may develop following exposure to air pollutants as heavy metals impair enzymatic functions and signaling processes within the cell, generating ROS, and mediating autoimmune responses. Research has documented deficiency in the immune system against ROS and impaired homeostasis, thereby increasing susceptibility to oxidative stress and heavy metal-related consequences (*Jafari et al., 2017*; *Zaky, 2017*). A few studies also depicted the potential of heavy metals to accumulate in the central neuronal system among ASD patients as a result of a significant decline in the ability to remove these toxic substances, culminating in neurotoxicity.

Exposure to neurotoxic heavy metals such as manganese (Mn), Cd, and Pb damages the developing children's brains (*Ijomone et al., 2020*). Exposure to heavy metals interferes with the translocation of copper (Cu) movement *via* the blood–brain barrier, which may affect the nervous system by facilitating oxidative stress-like situations and leading to ASD progression (*Liu et al., 2023*). Additionally, exposure to arsenic (As) has a substantial impact on the morphology of children's brains, resulting in a decline in cognitive function, including attention, comprehension, and language skills, and lower IQ scores (*Bjorklund et al., 2018*). *Skalny et al. (2018)* also reported that selenium (Se) deficiency causes brain imbalance and contributes to metabolic and psycho-metabolic issues in ASD (*Skalny et al., 2018*).

In Malaysia, earlier studies have reflected a high risk of exposure to heavy metals among children in Kuala Lumpur (*Kasmuri, 2020*; *Abdul Hamid et al., 2020*). A study conducted

two decades ago found approximately 11.7% of children in urban areas had blood Pb level (BLL) of more than 10.0 g/dL (*Hashim et al., 2000*)), which is greater than the standard recommended limits (*Curry et al., 2019*). Heavy metals measurement of classroom dust in Kuala Lumpur revealed strong evidence of exposure to several heavy metals among children within the area, with a positive correlation between Cu exposure and respiratory effect (*Tan et al., 2018*). An indication of environmental exposure can also be gleaned from studies reporting heavy metals (Cd, Pb, As, Mn) contamination in Kuala Lumpur's water supply (*Mohamad, Isa & Ajid, 2022*; *Mohd Hasni et al., 2017*; *Kasmuri, 2020*; *Abdul Hamid et al., 2020*) and ambient air (*Ismail et al., 2019*; *Wahab et al., 2020*). In the latter studies, the levels of Pb and Cd were greater than the safe level (*Ambient Air Quality Criteria. MECP T, ON, Canada, 2020*). In the same vein, a study on surface soil in Kuala Lumpur urban areas reported a high amount of heavy metal Cd in the samples, with potential health effects (*Abdul Hamid et al., 2020*).

Nevertheless, there is a dearth of toxicological studies on the impact of heavy metals like Mn, Pb, Cd, As, and Se on neurodevelopmental disorders in young children, particularly ASD. While numerous toxicological studies have been conducted in developed countries to assess the role of heavy metals on neurobehavioural disorders and human health risks, toxicological assessment in Malaysia is very limited, and further exploration is required to elucidate the association between these heavy metals and ASD among Malaysian children. This study aimed to assess the role of heavy metals (Pb, Cd, Mn, As, and Se) among Malaysian children in Kuala Lumpur.

## MATERIALS & METHODS

### Study design and study area

This study entails an unmatched case-control design conducted from December 15, 2020, until June 15, 2020. Preschoolers in the Federal Territory of Kuala Lumpur between the ages of 3 to 6 were recruited as the subjects (*Abd Wahil, Ja'afar & Isa, 2023*). In addition to being Malaysia's capital, Kuala Lumpur is a federal territory as in Fig. 1 (*Selangor, 2021*). It is the nation's biggest city. Kuala Lumpur and its surrounding areas host various industries, including manufacturing, mining, and processing. High traffic volumes in Kuala Lumpur also lead to high emissions from vehicles (*Wahab et al., 2020*).

### Sampling method and study population

Systematic random sampling was applied in enrolling ASD children at the national autism rehabilitation center in Kuala Lumpur (GENIUS KURNIA). Meanwhile, children without autism were recruited from public preschools (TASKA KEMAS) in Kuala Lumpur and allocated to the control group. With the approval of the institutions and related ministries, the two groups were randomly chosen from a name list of students using Microsoft Excel software. A total of 155 children (ASD = 81 and Control = 74 children) participated in the study as previously described in *Abd Wahil, Ja'afar & Isa (2022)*.

### Eligibility criteria

For both groups, the following conditions were used as exclusion criteria for recruitment (1) congenital anomaly or syndrome, (2) other neurodevelopmental or neurobehavioural

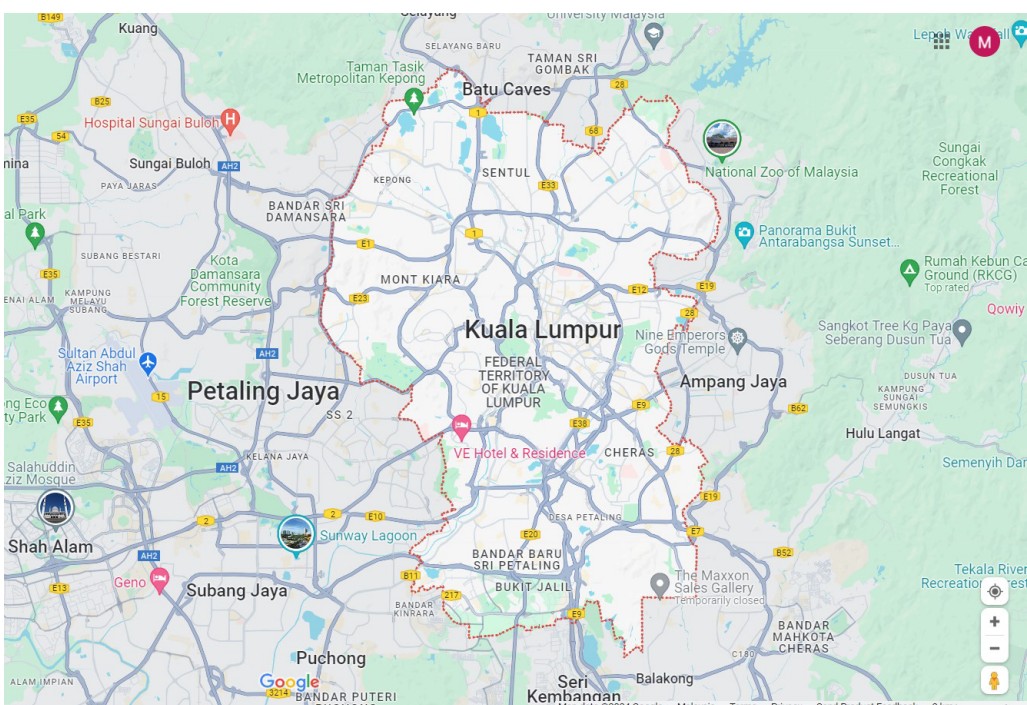

**Figure 1** **Map showing the study area, Kuala Lumpur, Malaysia.** Map Source: Google Maps.

disorders aside from autism or ASD, (3) endocrine disorders, (4) recent acute infections, surgical procedures, or trauma, and (5) those who were taking regularly oral medications, chemotherapy infusions, or chelation therapy for heavy metal removal. The inclusion criteria for this study are (1) Malaysian citizens (2) children aged between 3 and 6 years. (3) For the ASD group, the respondent was diagnosed clinically by the pediatrician of the Ministry of Health hospitals using DSM-5 criteria and International Classification of Disease (ICD-10). (4) For the control group, the children verified thriving by pediatrician confirmed that the patient lacked known ASD traits based on the Modified Checklist for Autism in Toddlers (M-CHAT) screening and the routine children's health examination at the 18- and 36-month mark (*Abd Wahil, Ja'afar & Isa, 2022*).

## Ethical approval

The National University of Malaysia (UKM) Research and Ethics Committee and the Medical Research (JEP 2019 –170) and Ethics Committee of the Ministry of Health (MOH) Malaysia (NMRR-18-3765-45117) have both given their approval to the study protocol. Before participating in the study, parents or legal guardians were provided with informed and written consent using a manual written form, and participation was optional.

## Research instruments and administration of the questionnaire

Through phone calls, text messages, and emails, the researcher requested that each participant's parent, complete the self-administered online survey using a Google Form. This method of online information gathering was preferred for several benefits,

including easy accessibility by phone or computer, user-friendliness, convenience at any time particularly for working parents, better data management and record keeping, confidentiality, and data analysis. In addition, the research approach aligns with the preventive measures against COVID-19 transmission through close contact during the epidemic years in Malaysia. Knowledge assessment on Pb exposure among children was conducted using a validated Malay-version questionnaire (*Abd Wahil, Ja'afar & Isa, 2022*). The children's dietary pattern was assessed using non-validated questions related to the source-receptor exposure pathway.

## Measurement of anthropometric data and analysis of urine samples

Data were collected as previously described in *Abd Wahil, Ja'afar & Isa (2023)*. Specifically, the child's anthropometric data, such as height and weight, were assessed using calibrated digital weighing scales (Omron; Kyoto, Japan) equipped with a height measurement stand. A standard reagent was used for the calibration but no validation test was performed.

Measurements were taken and recorded by the researcher in the classroom. Each participant's parent was provided with a sterile polyethene urine container that had been pre-treated with 20.0% nitric acid ($HNO_3$) solution and rinsed with deionized water. At the same time, instructions on clinical sample collection were explained by the researcher which included (1) the urine collection should be clean, the urine sample volume should be between 5.0 and 10.0 mL, (2) foreign substances such as body soap or detergent should be absent in the sterile urine container, (3) the urine sample should not be diluted with water, and (4) the urine container and biohazard zip bag should be properly closed. The children were permitted to eat and drink as usual. Parents delivered the urine samples to the researcher upon highlighting their children at the institutions, The urine samples were code-labeled and delivered to an environmentally accredited laboratory at the Faculty of Science and Technology, National University of Malaysia, Bangi, Selangor, within 24 h and kept at −22.0 °C. The urine samples were prepared in the lab setting by combining one mL of a urine sample with 10 mL of a 0.2% nitric acid ($HNO_3$) solution at a ratio of 1:10. This sample preparation is crucial to facilitate the digestion process for the removal organic matter in the clinical samples. Inductively coupled plasma mass spectrometry (ICPMS) (Perkin Elmer, Waltham, MA, USA) was used to analyze the prepared urine samples.

## Data analysis

The IBM Statistical Package for Social Sciences (SPSS) software (version 26, IBM, Chicago, IL, USA) was used to evaluate the dataset from the questionnaire and the laboratory ICPMS results. Before performing statistical analysis, the data normality was investigated graphically (using a histogram and a Q-Q plot) and statistically (using skewness, kurtosis, and Shapiro–Wilks/Kolmogorov–Smirnov statistics). For the participants' demographic characteristics, the frequencies and percentages were presented. The Mann–Whitney U test (for non-normal distribution) was used to analyze the group differences in Pb, Cd, Mg, As, and Se levels, and to evaluate the potentially related risk factors (quantitative variables). The categorical variables were reported in frequencies and column percentages following Chi-square and Yates correction for the continuity test. To evaluate the factors
(independent variables included heavy metals) associated with ASD, multiple logistic regression analyses were conducted. An estimation of the impact (odds ratio) of the components was obtained based on the final prediction model. In this investigation, a *p*-value of 0.05 or lower was considered statistically significant.

## RESULTS

### Descriptive results

The result of this study is related to our earlier published study and not independent of each other (*Abd Wahil, Ja'afar & Isa, 2022*; *Abd Wahil, Ja'afar & Isa, 2023*). Tables 1 and 2 depict the general characteristics between the cases and the control group. The male-to-female ratio was about 5:1 and 1:1 for children with ASD and no autism, respectively. Approximately 17.3% of children with ASD were able to talk by the time they turned 3 years old, despite having the disorder. Since the ability to talk was a requirement for inclusion in the control group, all children who did not have autism (100.0%) could talk by the time they turned 3 years old (Table 1).

As shown in Table 3, 81.1% of parents of children who did not have autism and 65.4% of parents of children with ASD did not smoke, respectively ($p = 0.011$). In the groups of children with ASD (93.8%) and without autism (82.4%), the parents reported no risk of heavy metals exposure at work ($p = 0.027$). Parental gender, child age, immunization status, child BMI, family history of ASD, obstetric risk factors, location of the home (next to a main road, industry, or construction site), and source of drinking water did not differ significantly between the groups ($p > 0.05$) (Table 3). Within the ASD group, those living outside Kuala Lumpur were relatively higher compared to those living in Kuala Lumpur, and vice versa for the TD group (*Abd Wahil, Ja'afar & Isa, 2022*).

More than half of the children with ASD were the first child in the household, whereas 30.0% of the children with normal development were the second child ($p < 0.001$). Most parents ($p < 0.001$) had secondary education and came from the B40 income category, which had monthly incomes of less than RM 5000. The majority of ASD kids reside primarily in Selangor, which is outside of Kuala Lumpur (*Abd Wahil, Ja'afar & Isa, 2022*). Both groups' offspring were mostly Malay ($p = 0.003$).

### Levels of heavy metals in the urine samples

Table 4 depicts the laboratory analysis of urinary Pb, Cd, As, Mg, and Se between the ASD group and control group. The urinary levels of Pb ($0.19 \pm 0.16 \, \mu g/dL$ *vs* $0.46 \pm 0.51 \, \mu g/dL$), Cd ($0.04 \pm 0.04$) mcg/dL *vs* $0.11 \pm 0.07 \, \mu g/dL$), and As ($3.80 \pm 3.40$ mcg/dL *vs* $6.91 \pm 8.22$ mcg/dL) were significantly lower ($p < 0.05$) in ASD children compared to children without autism. However, no statistical difference was observed between the ASD group and control group for Mn ($p = 0.659$) and Se ($p = 0.875$). The majority of children in both groups (91% for As, 98% for Cd, and 90% for Pb) had urine heavy metals lower than 15.1 mcg/dL, 1.0 μg/dL, and 1.0 mcg/dL, respectively which are the minimal risk value for noncarcinogenic detrimental human health effect due to the heavy metal's exposure (*ATSDR, 2007*; *ATSDR, 2020*; *Faroon et al., 2012*). The result of Pb in this result is similar to the reported Pb level finding in our previous article (*Abd Wahil, Ja'afar & Isa, 2022*).

**Table 1 Respondent characteristics.**

| Characteristics | Frequency (%) | |
|---|---|---|
| | ASD ($n = 81$) | TD ($n = 74$) |
| **Parents** | | |
| **Age group** | | |
| 30 years old and below | 1(1.2) | 14 (18.9) |
| More than 30 years old | 80(98.8) | 60 (81.1) |
| **Gender** | | |
| Male | 19 (23.5) | 22 (29.7) |
| Female | 62 (76.5) | 52 (70.3) |
| **Education level** | | |
| Primary education | 0 (0.0) | 0 (0.0) |
| Secondary education | 12 (14.8) | 45 (60.8) |
| Tertiary education | 69 (85.2) | 29 (39.2) |
| **Income classification** | | |
| B40 | 39 (48.1) | 58 (78.4) |
| M40 | 34 (42.0) | 16 (21.6) |
| T20 | 8 (9.9) | 0 (0.0) |
| **Residential area** | | |
| Kuala Lumpur | 31 (38.3) | 61 (82.4) |
| Outside Kuala Lumpur | 50 (61.7) | 13 (17.6) |
| **Children's background** | | |
| **Age group** | | |
| 4 years old and below | 5 (6.2) | 10 (13.5) |
| More than 4 years old | 76 (93.8) | 64 (86.5) |
| **Gender** | | |
| Male | 68 (84.0) | 39 (52.7) |
| Female | 13 (16.0) | 35 (47.3) |
| **Race** | | |
| Malay | 63 (77.8) | 70 (94.6) |
| Non-Malay | 18 (22.2) | 4 (5.4) |
| **Immunization status** | | |
| Up to date | 80 (98.8) | 70 (94.6) |
| Missed | 1 (1.2) | 4 (5.4) |
| **Speak at 3 years old** | | |
| Yes | 14 (17.3) | 74 (100.0) |
| No | 67 (82.7) | 0 (0.0) |
| **ASD among siblings** | | |
| Yes | 7 (8.6) | 4 (5.4) |
| No | 74 (91.4) | 70 (94.6) |

**Table 1** (*continued*)

| Characteristics | Frequency (%) | |
|---|---|---|
| | ASD (*n* = 81) | TD (*n* = 74) |
| **Maternal obstetric background** | | |
| **Advanced maternal age** | | |
| 35 years old and below | 78 (96.3) | 70 (94.6) |
| More than 35 years old | 3 (3.7) | 4 (5.4) |
| **Birth order** | | |
| First child | 46 (56.8) | 22 (29.7) |
| Subsequent child | 35 (43.2) | 52 (70.3) |
| **GDM** | | |
| Yes | 17 (21.0) | 19 (25.7) |
| No | 64 (79.0) | 55 (74.3) |
| **PIH** | | |
| Yes | 2 (2.5) | 3 (4.1) |
| No | 79 (97.5) | 71 (95.9) |
| **Anemia in pregnancy** | | |
| Yes | 27 (33.3) | 19 (25.7) |
| No | 54 (66.7) | 55 (74.3) |
| **Other comorbidities during pregnancy** | | |
| Yes | 12 (14.8) | 8 (10.8) |
| No | 69 (85.2) | 66 (89.2) |
| **Prematurity** | | |
| Yes | 9 (11.1) | 11 (14.9) |
| No | 72 (88.9) | 63 (85.1) |
| **Place of birth** | | |
| Government hospital | 42 (51.9) | 63 (85.1) |
| Private hospital | 39 (48.1) | 11 (14.9) |
| **State of birth** | | |
| Kuala Lumpur | 30 (37.0) | 42 (56.8) |
| Outside Kuala Lumpur | 51 (63.0) | 32 (43.2) |
| **Mode of delivery** | | |
| Spontaneous vertex | 46 (56.8) | 50 (67.6) |
| Assisted delivery | 11 (13.6) | 3 (4.0) |
| Cesarean section | 24 (29.6) | 21 (28.4) |
| **Birth weight** | | |
| 2,500 g and below | 8 (9.9) | 9 (12.2) |
| 2,501 g until 4,000 g | 71 (87.6) | 64 (86.5) |
| More than 4,000 g | 2 (2.5) | 1 (1.3) |
| **Neonatal complication** | | |
| Yes | 14 (17.3) | 9 (12.2) |
| No | 67 (82.7) | 65 (87.8) |
| **Breastfeeding** | | |
| Yes | 79 (97.5) | 73 (98.6) |
| No | 2 (2.5) | 1 (1.4) |

 

**Table 1** (*continued*)

| Characteristics | Frequency (%) | |
|---|---|---|
| | ASD (*n* = 81) | TD (*n* = 74) |
| **Duration of breastfeeding** | | |
| 6 months and below | 28 (34.6) | 28 (37.8) |
| 7 months until 24 months | 41 (50.6) | 34 (46.0) |
| More than 24 months | 12 (14.8) | 12 (16.2) |
| **Environmental exposure background** | | |
| **Type of the house** | | |
| Bungalow house | 5 (6.2) | 5 (6.8) |
| Semi-detached house | 5 (6.2) | 1 (1.4) |
| Terrace house | 26 (32.1) | 18 (24.3) |
| Condominium | 26 (32.1) | 7 (9.5) |
| Apartment | 2 (2.5) | 3 (4.0) |
| Flat house | 17 (20.8) | 40 (54.0) |
| **Age of the house** | | |
| 25 years and below | 64 (79.0) | 47 (63.5) |
| 26–45 years | 17 (21.0) | 23 (31.1) |
| More than 45 years | 0 (0.0) | 4 (5.4) |
| **House near the main road** | | |
| Yes | 48 (59.3) | 41 (55.4) |
| No | 33 (40.7) | 33 (44.6) |
| **House near the factory** | | |
| Yes | 9 (11.1) | 11 (14.9) |
| No | 72 (88.9) | 63 (85.1) |
| **House near construction site** | | |
| Yes | 15 (18.5) | 18 (24.3) |
| No | 66 (81.5) | 56 (75.7) |
| **Parental smoking status** | | |
| Active smoker | 12 (14.8) | 11 (14.9) |
| Ex-smoker | 16 (19.8) | 3 (4.1) |
| Non-smoker | 53 (65.4) | 60 (81.0) |
| **Parental risk at the workplace** | | |
| Yes | 5 (6.2) | 13 (17.6) |
| No | 76 (93.8) | 61 (82.4) |
| **Exposure to the soil** | | |
| Everyday | 3 (3.7) | 5 (6.8) |
| Once a week | 15 (18.5) | 22 (29.7) |
| Once a month | 33 (40.8) | 26 (35.1) |
| Never | 30 (37.0) | 21 (28.4) |
| **Sucking own hand** | | |
| All the time | 7 (8.6) | 2 (2.7) |
| Just before sleep | 14 (17.3) | 5 (6.8) |
| Never suck hand | 60 (74.1) | 67 (90.5) |

**Table 1** (*continued*)

| Characteristics | Frequency (%) | |
|---|---|---|
| | ASD (*n* = 81) | TD (*n* = 74) |
| **Washing hand practice** | | |
| Frequent | 59 (72.8) | 68 (91.9) |
| Sometimes | 15 (18.5) | 6 (8.1) |
| Seldom | 7 (8.6) | 0 (0.00) |
| Dietary habits of children | | |
| **Drinking water from tap water** | | |
| Yes | 75 (92.6) | 66 (89.2) |
| No | 6 (7.4) | 8 (10.8) |
| **PICA** | | |
| Yes | 16 (19.8) | 4 (5.4) |
| No | 47 (58.0) | 61 (82.4) |
| Not sure | 18 (22.2) | 9 (12.2) |
| **Vitamin consumption** | | |
| Yes | 47 (58.0) | 42 (56.8) |
| No | 34 (42.0) | 32 (43.2) |
| **Drinking milk** | | |
| More than 2 times per day | 22 (27.2) | 17 (23.0) |
| 1 to 2 times per day | 33 (40.7) | 38 (51.3) |
| Never drink | 26 (32.1) | 19 (25.7) |
| **Eating Fruits** | | |
| Every meal | 10 (12.4) | 17 (23.0) |
| Once a day | 21 (25.9) | 19 (25.7) |
| Once a week | 27 (33.3) | 35 (47.3) |
| Never eat | 23 (28.4) | 3 (4.0) |
| **Eating vegetables** | | |
| Every meal | 16 (19.8) | 12 (16.2) |
| Once a day | 21 (25.9) | 20 (27.1) |
| Once a week | 12 (14.8) | 18 (24.3) |
| Never eat | 32 (39.5) | 24 (32.4) |

**Notes.**
There was no significant difference ($p < 0.05$) in any of the characteristics between the groups.
PICA, Eating nonfood items; PIH, Pregnancy-induced hypertension.

## Factors associated with ASD among the study population

Specific factors were identified to increase the odds of having an ASD child among the study population as shown in Table 5. Parental education, children's birth order, and gender were identified as significant factors of ASD as reported in our published study (*Abd Wahil, Ja'afar & Isa, 2022*; *Abd Wahil, Ja'afar & Isa, 2023*). Parents with tertiary education had 13.8 times higher odds of having an ASD child compared to parents with secondary education (adjusted odds ratio (aOR) = 13.8, 95% CI [3.6–53.8], $p < 0.001$). The odds of ASD in firstborn children were 6.5 times higher than in subsequent-born children (aOR = 6.5, 95% CI [1.8–23.5], $p = 0.004$). Likewise, the odds of ASD in male children were 9.5 times higher than for female children (aOR = 9.5, 95% CI [2.4–37.9], $p < 0.001$).

**Table 2  Respondent characteristics.**

| Characteristics | Mean (±SD) | |
| --- | --- | --- |
| | ASD ($n = 81$) | TD ($n = 74$) |
| **Parent's background** | | |
| Age (years) | 36.7 (±3.88) | 35.2 (±5.45) |
| Monthly income (RM) | 6699.9 (±3856.52) | 3960.0 (±2025.19) |
| **Children's background** | | |
| Birth weight (g) | 3.0 (±0.47) | 3.0 (±0.59) |
| Age (years) | 5.6 (±0.60) | 5.5 (±0.83) |
| Duration of breastfeeding (months) | 14.7 (±12.19) | 14.7 (±11.17) |
| BMI | 16.3 (±3.13) | 15.7 (±3.70) |
| **Maternal obstetric background** | | |
| Maternal age at pregnancy | 29.9 (±3.41) | 29.3 (±4.82) |
| The hb level at 36 weeks gestation | 11.6 (±1.22) | 11.8 (±2.89) |
| Gestational age at birth (week) | 37.9 (±2.51) | 38.3 (±1.69) |
| **Environmental exposure background** | | |
| Age of the house (years) | 22.3 (±14.62) | 18.3 (±9.78) |

Notes.
There was no significant difference ($p > 0.05$) in any of the characteristics between the groups.

## DISCUSSION

ASD is a neurobehavioural condition that stems from the interaction between environmental and genetic factors (*Masini et al., 2020*). As neurotoxicants, heavy metals play a crucial role in the development and onset of ASD by impairing the physiological function of organs and altering their morphological characteristics even at a low concentration (*Masini et al., 2020*). Young children are more susceptible to experiencing the impacts of neurotoxicants than adults given their organs' low capacity to detoxify harmful chemicals in the body (*Abd Wahil, Ja'afar & Isa, 2023*).

The present research findings refute the earlier hypotheses that high Pb, Cd, As, Mn, and low Se urine levels are associated with ASD in preschoolers in Urban areas of Kuala Lumpur. Specifically, Pb, Cd, and As urine levels were surprisingly greater in children without autism compared to ASD children. Cd content in urine samples from ASD children was also considerably lower than those without autism. Both groups of children had urine Pb concentrations below the permissible limit of 5.0 µg/dL as the highest level of urine Pb was 2.5 µg/dL. Most of the children (90.0%, n = 135/155) had urine Pb levels <1.0 µg/dL. Despite most of the hypotheses were not supported by the research findings, the levels of heavy metals in the urine samples of the study population still raise an important concern. This is because heavy metals are still toxic even in minute amounts, and harmful to children given their vulnerability. No safe level of these heavy metals is established to date and the underlying mechanisms are highly complex.

In terms of urine Cd levels, the result from this study is consistent with the reports from India (*Sehgal et al., 2019*) and China (*Li et al., 2022*) in which children with ASD demonstrated significantly lower blood Cd levels compared to the healthy control groups.

**Table 3 Bivariate relationship between preschoolers' sociodemographic characteristics among ASD and TD groups.**

| Variables | ASD($n = 81$) N (%) | TD($n = 74$) N (%) | 2-value | *p*-value |
|---|---|---|---|---|
| **Parent background** | | | | |
| **Age group** | | | | |
| 30 years old and below | 1 (1.2) | 14 (18.9) | 13.83 | <0.001[a] |
| More than 30 years old | 80 (98.8) | 60 (81.1) | | |
| **Gender** | | | | |
| Male | 19 (23.5) | 22 (29.7) | 0.78 | 0.376 |
| Female | 62 (76.5) | 52 (70.3) | | |
| **Education level** | | | | |
| Primary education | 0 (0.0) | 0 (0.0) | 35.19 | <0.001[a] |
| Secondary education | 12 (14.8) | 45 (60.8) | | |
| Tertiary education | 69 (85.2) | 29 (39.2) | | |
| **Income classification** | | | | |
| B40 | 39 (48.1) | 58 (78.4) | 17.92 | <0.001[a] |
| M40 | 34 (42.0) | 16 (21.6) | | |
| T20 | 8 (9.9) | 0 (0.0) | | |
| **Residential area** | | | | |
| Kuala Lumpur | 31 (38.3) | 61 (82.4) | 31.26 | <0.001[a] |
| Outside Kuala Lumpur | 50 (61.7) | 13 (17.6) | | |
| **Children's background** | | | | |
| **Gender** | | | | |
| Male | 68 (84.0) | 39 (52.7) | 17.66 | <0.001[a] |
| Female | 13 (16.0) | 35 (47.3) | | |
| **Age group** | | | | |
| 4 years old and below | 5 (6.2) | 10 (13.5) | 2.38 | 0.123 |
| More than 4 years old | 76 (93.8) | 64 (86.5) | | |
| **Race** | | | | |
| Malay | 63 (77.8) | 70 (94.6) | 8.98 | 0.003[a] |
| Non-Malay | 18 (22.2) | 4 (5.4) | | |
| **Immunization status** | | | | |
| Up to date | 80 (98.80) | 70 (94.60) | 1.026[#] | 0.311 |
| Missed | 1 (1.20) | 4 (5.40) | | |
| **Speak at 3 years old** | | | | |
| Yes | 14 (17.3) | 74 (100.0) | 107.81 | <0.001[a] |
| No | 67 (82.7) | 0 (0.0) | | |
| **ASD among siblings** | | | | |
| Yes | 7 (8.6) | 4 (5.4) | 0.61 | 0.433 |
| No | 74 (91.4) | 70 (94.6) | | |
| **Advanced maternal age** | | | | |
| 35 years old and below | 78 (96.3) | 70 (94.6) | 0.01 | 0.903 |
| More than 35 years old | 3 (3.7) | 4 (5.4) | | |

**Table 3** (*continued*)

| Variables | ASD($n = 81$) N (%) | TD($n = 74$) N (%) | 2-value | *p*-value |
|---|---|---|---|---|
| **GDM** | | | | |
| Yes | 17 (21.0) | 19 (25.7) | 0.48 | 0.490 |
| No | 64 (79.0) | 55 (74.3) | | |
| **PIH** | | | | |
| Yes | 2 (2.5) | 3 (4.1) | 0.01[#] | 0.918 |
| No | 79 (97.5) | 71 (95.9) | | |
| **Anaemia** | | | | |
| Yes | 27 (33.3) | 19 (25.7) | 1.09 | 0.297 |
| No | 54 (66.7) | 55 (74.3) | | |
| **Other comorbidities during pregnancy** | | | | |
| Yes | 12 (14.8) | 8 (10.8) | 0.55 | 0.458 |
| No | 69 (85.2) | 66 (89.2) | | |
| **Prematurity** | | | | |
| Yes | 9 (11.1) | 11 (14.9) | 0.48 | 0.486 |
| No | 72 (88.9) | 63 (85.1) | | |
| **Mode of delivery** | | | | |
| Spontaneous vertex | 46 (56.8) | 50 (67.6) | 4.63 | 0.099 |
| Assisted delivery | 11 (13.6) | 3 (4.0) | | |
| Cesarean section | 24 (29.6) | 21 (28.4) | | |
| **Birth weight** | | | | |
| 2,500 g and below | 8 (9.9) | 9 (12.2) | 0.45 | 0.800 |
| 2,501 g until 4,000 g | 71 (87.6) | 64 (86.5) | | |
| More than 4,000 g | 2 (2.5) | 1 (1.3) | | |
| **Neonatal complication** | | | | |
| Yes | 14 (17.3) | 9 (12.2) | 0.80 | 0.370 |
| No | 67 (82.7) | 65 (87.8) | | |
| **Breastfeeding** | | | | |
| Yes | 79 (97.5) | 73 (98.6) | 0.000[#] | 1.000 |
| No | 2 (2.5) | 1 (1.4) | | |
| **Duration of breastfeeding** | | | | |
| 6 months and below | 28 (34.6) | 28 (37.8) | 0.34 | 0.845 |
| 7 months until 24 months | 41 (50.6) | 34 (45.9) | | |
| More than 24 months | 12 (14.8) | 12 (16.2) | | |
| **Birth Order** | | | | |
| First child | 46 (56.8) | 22 (29.7) | 0.57 | <0.001[a] |
| Subsequent child | 35 (43.2) | 52 (70.3) | | |
| **Place of birth** | | | | |
| Government hospital | 42 (51.9) | 63 (85.1) | 19.60 | <0.001[a] |
| Private hospital | 39 (48.1) | 11 (14.9) | | |

**Table 3** (*continued*)

| Variables | ASD(*n* = 81)<br>N (%) | TD(*n* = 74)<br>N (%) | 2-value | *p*-value |
|---|---|---|---|---|
| **State of birth** | | | | |
| Kuala Lumpur | 30 (37.0) | 42 (56.8) | 6.05 | 0.014[a] |
| Outside Kuala Lumpur | 51 (63.0) | 32 (43.2) | | |
| **Environmental exposure background** | | | | |
| **Age of the house** | | | | |
| 25 years and below | 64 (79.0) | 47 (63.5) | 8.75 | 0.013[a] |
| 26–45 years | 17 (21.0) | 23 (31.1) | | |
| More than 45 years | 0 (0.0) | 4 (5.4) | | |
| **House near the main road** | | | | |
| Yes | 48 (59.3) | 41 (55.4) | 0.24 | 0.628 |
| No | 33 (40.7) | 33 (44.6) | | |
| **House near the factory** | | | | |
| Yes | 9 (11.1) | 11 (14.9) | 0.49 | 0.486 |
| No | 72 (88.9) | 63 (85.1) | | |
| **House near construction site** | | | | |
| Yes | 15 (18.50) | 18 (24.30) | 0.78 | 0.378 |
| No | 66 (81.50) | 56 (75.70) | | |
| **Parental smoking status** | | | | |
| Active smoker | 12 (14.8) | 11 (14.9) | 9.07 | 0.011[a] |
| Ex-smoker | 16 (19.8) | 3 (4.1) | | |
| Non-smoker | 53 (65.4) | 60 (81.1) | | |
| **Parental risk at the workplace** | | | | |
| Yes | 5 (6.2) | 13 (17.6) | 4.89 | 0.027[a] |
| No | 76 (93.8) | 61 (82.4) | | |
| **Exposure to the soil** | | | | |
| Everyday | 3 (3.7) | 5 (6.8) | 3.95 | 0.267 |
| Once a week | 15 (15.8) | 22 (29.7) | | |
| Once a month | 33 (40.7) | 26 (35.1) | | |
| Never | 30 (37.0) | 21 (28.4) | | |
| **Vitamin consumption** | | | | |
| Yes | 47 (58.0) | 42 (56.8) | 1.37 | 0.505 |
| No | 34 (42.0) | 32 (43.2) | | |
| **Drinking milk** | | | | |
| More than 2 times per day | 22 (27.2) | 17 (23.0) | 1.77 | 0.413 |
| 1 to 2 times per day | 33 (40.7) | 38 (51.4) | | |
| Never drink | 26 (32.1) | 19 (25.7) | | |
| **Eating fruits** | | | | |
| Every meal | 10 (12.3) | 17 (23.0) | 18.05 | <0.001[a] |
| Once a day | 21 (25.9) | 19 (25.7) | | |
| Once a week | 27 (33.3) | 35 (47.3) | | |
| Never eat | 23 (28.4) | 3 (4.1) | | |

**Table 3** (*continued*)

| Variables | ASD(*n* = 81) N (%) | TD(*n* = 74) N (%) | 2-value | *p*-value |
|---|---|---|---|---|
| **Eating vegetables** | | | | |
| Every meal | 16 (19.8) | 12 (16.2) | 2.63 | 0.453 |
| Once a day | 21 (25.9) | 20 (27.0) | | |
| Once a week | 12 (14.8) | 18 (24.3) | | |
| Never eat | 32 (39.5) | 24 (32.4) | | |
| **Child's BMI** | 15.6(3.12) | 14.79(3.19) | 3,016.5[c] | 0.082[b] |
| **The Hb level at 36 weeks gestation** | 11.6(1.3) | 11.65(1.4) | 3,482.5[c] | 0.944[b] |

**Notes.**
2-value = Chi-square statistical value
[a]*p* < 0.05 indicates a significant statistical result.
[b]Mann Whitney U Test, median (IQR) as a descriptive result.
[c]Z-statistical value.
[#]Yates continuity for correction test.
Income classification: B40 represents the bottom 40% of income earners (net income below RM 4,850), M40 represents the middle 40% of income earners (net income of RM 4,851 to RM 10,960), and T20 constitutes the 20% top income earners (net income greater than RM 10,960) (Department of Statistics (DOSM), 2022).

**Table 4** Comparisons of urinary Pb, Cd, As, Mg, and Se levels between the ASD and TD groups.

| Urinary heavy metals | Group | Sample | Mean | SD | Median | Interquartile range | Difference in median/ratio of ASD/TD | Statistical value (z) | *p*-value[a] |
|---|---|---|---|---|---|---|---|---|---|
| **Overall (*n* = 155)** | | | | | | | | | |
| Pb (μg/dL) | ASD | 81 | 0.26 | 0.31 | 0.19 | 0.11 | 0.41 | 5,059 | <0.001[#] |
| | TD | 74 | 0.58 | 0.41 | 0.46 | 0.51 | | | |
| Cd (mcg/dL) | ASD | 81 | 0.04 | 0.03 | 0.04 | 0.04 | 0.36 | 5,450 | <0.001[#] |
| | TD | 74 | 0.20 | 0.36 | 0.11 | 0.07 | | | |
| Mn (mcg/dL) | ASD | 81 | 0.24 | 0.23 | 0.17 | 0.30 | 0.94 | 3,802 | 0.659 |
| | TD | 74 | 0.41 | 0.86 | 0.18 | 0.18 | | | |
| As (mcg/dL) | ASD | 81 | 5.32 | 5.12 | 3.78 | 3.40 | 0.54 | 3,120 | 0.004[#] |
| | TD | 74 | 9.76 | 12.20 | 6.91 | 8.22 | | | |
| Se (μg/dL) | ASD | 81 | 7.13 | 3.63 | 6.91 | 5.61 | 0.96 | 3,041 | 0.875 |
| | TD | 74 | 7.33 | 3.98 | 7.20 | 7.08 | | | |

**Notes.**
[#]*p* < 0.05 indicates a significant statistical result.
[a]Mann Whitney U Test.

A recent systematic review and meta-analysis also found that the ASD group had lower urine Cd concentrations, which reflects the lower capacity of ASD patients to excrete heavy metals (*Ding et al., 2023*). In contrast, systematic reviews of several studies revealed that Cd in both hair and urine samples were adversely and positively associated with ASD children (*Sulaiman, Wang & Ren, 2020*). *Skogheim et al. (2021)* found that individuals exposed to Cd experienced a significantly increased risk of having an ASD offspring. Moreover, higher Cd concentrations were observed in Asian and European patients, whereas the opposite result was observed in North American patients (*Ding et al., 2023*). These findings indicate geographical locations may contribute to the difference in Cd concentrations among ASD

**Table 5** Multiple logistic regression models for factors associated with having an ASD children among the study population.

| Variable | adjusted OR | 95% CI OR | Wald | *p*-value[#] | $R^2$ |
|---|---|---|---|---|---|
| Parent education background | | | | | |
| Tertiary education | 13.84 | (3.56, 53.75) | 14.41 | <0.001 | 0.79 |
| Secondary education | 1.00 | | | | |
| | | | | | |
| Children's birth order | | | | | |
| First child | 6.52 | (1.81, 23.48) | 8.22 | 0.004 | |
| Subsequent child | 1.00 | | | | |
| Children gender | | | | | |
| Male | 9.54 | (2.40, 37.85) | 10.28 | <0.001 | |
| Female | 1.00 | | | | |
| Heavy metal | | | | | |
| Pb level (µg/dL) | 0.24 | (0.05, 1.18) | 3.09 | 0.079 | |

**Notes.**

[#] $p < 0.05$ indicates a significant statistical result. No interaction between variables, no multicollinearity, and no influential outlier, both numerical variables were linear, Hosmer Lemeshow goodness of fit test was not significant ($p = 0.999$), Sensitivity, specificity of model-s prediction = 97.4%.

patients, including children. However, the specific event remains unclear and more research is required. Overall, the present results support the need to reduce Cd consumption and environmental exposure to this heavy metal, particularly in children whose brains are still developing.

The bivariate analysis also revealed that the urine Pb content from ASD children was significantly lower than those without autism. Pb is a non-ferrous heavy metal that exists naturally and is highly abundant in the environment. This toxic substance can cause irreversible neurotoxicity even at a very low concentration (*Abd Wahil, Ja'afar & Isa, 2022*). The findings from this study align and contradict with several studies reporting the Pb levels in the urine and blood samples of ASD patients, including children (*Rahbar et al., 2015*; *Rahbar et al., 2020*; *Alabdali, Al-Ayadhi & El-Ansary, 2014*; *Yorbik et al., 2010*). For instance, blood Pb levels were significantly lower in children with ASD compared to the control groups (*Rahbar et al., 2015*; *Rahbar et al., 2020*; *Alabdali, Al-Ayadhi & El-Ansary, 2014*), which is expected to reflect in the urine samples as well. This was evident in the studies by *Yorbik et al. (2010)* and *Nakhaee et al. (2022)* as Pb levels in urine samples from ASD children were much lower compared to children without autism. A meta-analysis also found that ASD patients from Asian and European countries had significantly higher blood and hair Pb levels, as well as lower urine Pb concentrations, than the healthy controls (*Ding et al., 2023*). However, several studies have contradicted the findings of this study regarding the level of Pb concentration in ASD children where there was a much higher level of the toxic heavy metal in their urine compared to the control groups (*Adams et al., 2013*; *Adams et al., 2017*; *Blaurock-Busch, Amin & Rabah, 2011*; *Metwally et al., 2015*; *Nabgha et al., 2020*). A meta-analysis also further strengthened the fact that urine Pb can be higher in ASD children compared to healthy control children (*Stojsavljević, Lakićević & Pavlović, 2023*). Although blood and hair samples were not analyzed in the present study, the findings

suggest that ASD children may have lower excretion ability for heavy metals, including Pb (*Cekici & Sanlier, 2019*). Previous studies posit that the declined ability might be reduced levels of anti-oxidants in ASD patients, coupled with the excretion of calcium ions which competitively the removal of Pb, leading to lower binding of Pb to the anti-oxidants and a low excretion rate of Pb *via* urine (*Kern et al., 2007*).

The Malaysian government's decision to phase out Pb from petrol beginning in early 1998 may be one of the factors behind the low Pb levels in children over the past 20 years. Consequently, between 1990 and 2004, the amount of Pb in the air significantly decreased. The impact of this policy can be observed by a study in Kuala Lumpur that proves blood Pb levels among children decreased from 5.26 g/dL in 2000 (*Hashim et al., 2000*) to 3.40 g/dL in 2007 (*Elias et al., 2007*). The Control of Lead Poisoning (Lead in Paint) Regulations 2019 were implemented in Malaysia and provide restrictions on the amount of lead that can be found in paints, coatings, and related items. The Malaysian government has also developed several regulations, such as the most recent National Automotive Policy (NAP) 2020, to promote the usage of alternative vehicles. These include public transit (such as buses, monorails, and electric trains) and battery-electric vehicles (BEVs). Under Regulation 28 of the Food Regulations of 1985, Malaysia also has the authority to impose restrictions on the importation of ceramic ware that releases no more Pb or Cd than is necessary. Cd and its compounds are listed as scheduled wastes under the Environmental Quality (Scheduled Wastes) Regulations of 2005, which places strict limits on their management and disposal.

Regarding As levels, this study found lower urine As concentrations in ASD children compared to the control group, which corroborates the results from research conducted in other countries (*Skalny et al., 2017*; *Rahbar et al., 2020*). Accordingly, 34% of children aged 2–4 years and affected with ASD had lower As concentrations in their hair metal than those without autism (*Skalny et al., 2017*). In Jamaica, *Rahbar et al. (2020)* found that ASD children had considerably lower blood As concentrations than the control group, which is consistent with the present study. Subgroup analysis following a meta-analysis also reported higher urine As concentrations in ASD children relative to healthy controls (*Ding et al., 2023*). Another review also found that urine As concentration is much higher in ASD children compared to healthy control children (*Shiani et al., 2023*). This information further validates the hypothesis that the development of ASD may be associated with exposure to As. As may affect cellular metabolic processes, ASD was once described as a metabolic disorder characterised by alterations in the metabolism of glucose, amino acids and lipids (*Gevezova et al., 2021*; *Du et al., 2022*). On the other hand, some studies reported no significant association between exposure to As *via* measurement in hair samples and the development of ASD in children (*De Palma et al., 2012*; *Fido & Al-Saad, 2005*). While the main reason for these discrepancies is not fully understood, geographical variation and differences in the samples analysed may play an important role in the results.

The low As level among children also can be explained by Malaysia's advocation of secure farming methods to reduce arsenic contamination in food crops. This entails teaching farmers about appropriate irrigation practices, discouraging the use of pesticides and fertilizers that contain arsenic, and regularly testing the soil to check for arsenic concentrations. Likewise, the allowable levels for As in food products have been defined

by Malaysia's guidelines for food safety. To ensure compliance and stop the distribution of food products with excessive arsenic levels, regular monitoring and inspections are carried out.

As for Se and Mn, no significant difference was observed between both groups. Specifically, Se which acts as an antioxidant was lower in ASD children compared to those without autism but the difference was not statistically significant. Nevertheless, the results suggest that ASD children may have a diminished capacity to excrete heavy metals due to low antioxidants, making them poor detoxifiers (*Holmes, Blaxill & Haley, 2003*; *Bradstreet et al., 2003*). Antioxidants function by several different methods, including scavenging free radicals, stopping radical chain reactions, and building stable complexes with heavy metals. As a result, children with ASD are more likely to have higher levels of heavy metals in their brains and less amounts will be excreted in urine due to the lower levels of antioxidants in their bodies (*Jan et al., 2015*).

Regression analysis revealed that specific factors such as parental education, children's birth order, and the children's gender were associated with having an ASD child. For instance, male children were nine times more likely to develop ASD compared to their female counterparts. Similar reports could be gleaned from previous studies conducted in other countries (*Baxter et al., 2015*; *Loomes, Hull & Mandy, 2017*). The underlying event for higher susceptibility to ASD among male children is not well understood, thus further research is required to elucidate the relationship. Additionally, firstborn children had a six times higher likelihood of developing ASD compared to children born later in the birth order. This finding may arise from the fact that parents might stop having children since their first child is affected by ASD. An earlier study also found that firstborn children of a couple had a threefold increased risk of ASD compared to children born later to mothers in their 20s and fathers in their 40s (*Durkin et al., 2008*).

In terms of educational qualification, parents with a university education in the present study had a 13 times greater chance of having an ASD child than parents with only secondary education, which aligns with a study conducted by (*Eow et al., 2020*). This finding may be linked to a better ability to recognize autism or more access to diagnosis among parents with higher educational qualifications, which may be less obtainable in less-educated parents. Mothers with higher education led to autistic children being identified and diagnosed because they had the necessary knowledge to recognize their child's developmental and behavioural abnormalities (*Bhasin & Schendel, 2007*). They also have the financial means to access the necessary medical care, which suggests the effects of social class on autism diagnosis (*Bhasin & Schendel, 2007*).

## LIMITATION

Despite the pertinent findings reported in this study, the limitations should be taken into account in the data interpretation. Complex pathogenic mechanisms caused by heavy metals occurring in the brain may not be fully explained in this study since these elements were only examined in urine samples. Additionally, the heterogeneity (spectrum) of ASD, the subjects' varied geographic locations, or methodological variations may contribute to

the relative inconsistency in the results, particularly in cases where ASD children recorded various levels of heavy metal. Nevertheless, these findings suggest more research is needed to examine the potential contribution of lead and other heavy metals to the development of ASD among children.

## Conclusions and recommendations for future research

Overall, this unmatched case-control study revealed low levels of heavy metals (Pb, Cd, As, Se and Mn) in the urine of most of the sampled children in Kuala Lumpur, irrespective of their ASD status. Children with ASD had significantly lower Pb, Cd, and As urine concentrations than those without autism, whereas the Se and Mn concentrations were similar in both groups. The low levels of heavy metals (Pb, Cd, and As) in ASD children's urine may stem from their diminished capacities to excrete heavy metals due to low antioxidants, making them poor detoxifiers. Nevertheless, heavy metals are cumulative toxins and few of them have a very long half-life in the body. In addition, the exposure of children to heavy metals even at low amounts may have long-term consequences. Specific factors were identified to increase the odds of having an ASD child in this study, which included firstborn male children, higher parental education levels, and male gender. These factors may be considered by the current antenatal and child development care, serving as a reminder and guidelines for healthcare professionals who are evaluating a child's development for the early detection of ASD. Further research on the assessment of heavy metals among ASD children in the country will be beneficial for future reference. Interventional studies and programmes, including awareness campaigns, may also be considered in future research to develop effective methods of addressing exposure to heavy metals at the community level.

## ACKNOWLEDGEMENTS

The Department of Community Health at UKM and the Faculty of Medicine were helpful in the conduct of this study, which the authors gratefully acknowledge.

### Funding
The authors received no funding for this work.

### Competing Interests
The authors declare there are no competing interests.

### Author Contributions
- Muhammad Ridzwan Rafi'i conceived and designed the experiments, performed the experiments, analyzed the data, prepared figures and/or tables, authored or reviewed drafts of the article, and approved the final draft.
- Mohd Hasni Ja'afar conceived and designed the experiments, performed the experiments, analyzed the data, prepared figures and/or tables, authored or reviewed drafts of the article, and approved the final draft.

- Mohd Shahrol Abd Wahil conceived and designed the experiments, performed the experiments, authored or reviewed drafts of the article, and approved the final draft.
- Shahrul Azhar Md Hanif analyzed the data, prepared figures and/or tables, authored or reviewed drafts of the article, and approved the final draft.

## Human Ethics

The following information was supplied relating to ethical approvals (i.e., approving body and any reference numbers):

The National University of Malaysia (UKM) Research and Ethics Committee and the Medical Research (JEP 2019 –170) and Ethics Committee of the Ministry of Health (MOH) Malaysia (NMRR-18-3765-45117) have both given their approval to the study protocol.

## Data Availability

The data is available in the Supplemental File.

## Supplemental Information

Supplemental information for this article can be found online at http://dx.doi.org/10.7717/peerj.17660#supplemental-information.

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
