# Peer review of "Urine manganese, cadmium, lead, arsenic, and selenium among autism spectrum disorder children in Kuala Lumpur"

_PeerJ, doi:10.7717/peerj.17660_

## Round 0.1 · original submission · Major Revisions

Dear authors, thank you for your submission. At this stage, we believe that improving the quality and impact of this work requires major revisions. Please, refer to the reviewers' comments for further details.

**Language Note:** PeerJ staff have identified that the English language needs to be improved. When you prepare your next revision, please either (i) have a colleague who is proficient in English and familiar with the subject matter review your manuscript, or (ii) contact a professional editing service to review your manuscript. PeerJ can provide language editing services - you can contact us at [email protected] for pricing (be sure to provide your manuscript number and title). – PeerJ Staff

Reviewer 1 ·

Basic reporting

This is an interesting paper comparing levels of several metals in children with ASD vs controls. The paper is well-organized and mostly well written, with a need for clarification of some points

Experimental design

The case-control design is appropriate.

Validity of the findings

The methodology seems valid.

See comments below

Additional comments

My major concern is that the discussion section focuses on studies which agree with the present results, but there are also studies that disagree which need to be mentioned, with a focus on systematic reviews.
Abstract: “always associated” should be “sometimes associated”
It is unclear what is meant by “normal values” since you have values for typical children.
It is unclear what is meant by “risk factors” – they should apply to specific measurements, such as level of lead.
Lines 34 and 35 contradict one another – were Mn and Se different in ASD or not?

Line 58: “more significantly” should be “in addition to” – both genetics and environment are important
Line 68 – clarify if you mean selenium deficiency or excess
Line 101 – exclusion 2 would eliminate all the ASD children
Table 1 and 2 – state p-values or clarify that none were significantly different
Table 3 – what is 2-value?
Table 3 – it states there were more females than males in the ASD group?
Define income classification

Table 4: since for several results the standard deviation is larger than the mean, this means the data is not normally distributed, so it is better to list the 1st and 3rd quartile in addition to the median. Add the % difference in medians between the two groups, or the ratio of ASD/TD.
Interpreting table 5:
Being more likely to being first born could be due to parents stopping having children after having one with ASD
Having higher parental education could result in better ability to recognize autism or access to diagnosis?

Within the ASD group, was there a difference in metal levels depending on living in Kuala Lumpur vs outside? Same question for the TD group.

Line 198 – what is “normal” levels – is this a laboratory reference range for adults?
Line 202 – no parents had only primary education, so are you comparing tertiary vs secondary?
Line 229 – the selenium levels were only slightly lower, and given the large standard deviation I would not say they were lower, but instead say they were not significantly different
The discussion focuses on studies where toxic metal levels were lower in ASD, but most studies have found it higher. Please give a balanced review, instead of only citing papers that agree with you. See this recent review for example: Association between heavy metals exposure (cadmium, lead, arsenic, mercury) and child autistic disorder: a systematic review and meta-analysis (nih.gov)
https://www.ncbi.nlm.nih.gov/pmc/articles/PMC10353844/pdf/fped-11-1169733.pdf

Reviewer 2 ·

Basic reporting

I want to thank the editor for the opportunity to evaluate the article and contribute as a Peerj magazine reviewer. The article entitled Urine manganese, cadmium, lead, arsenic, and selenium among autism spectrum disorder children in Kuala Lumpur is essential for the scientific community. It addresses a relevant topic: the assessment of metals involving autistic children. The assessment of metals in urine is not a new approach; however, when considering children with autism, this approach is strategic and essential for the scientific community. I saw this article receiving many readings and citations when it was published.
The article is well written and written in satisfactory English, but it needs necessary corrections before it can be published. I made a list of observations that require deeper corrections to improve the quality of the article.

Experimental design

The experimental project was well-designed and had ethical authorization for its execution. A total of 155 children were evaluated, 81 with autism and 74 without autism. Urine samples and questionnaires were collected for metal assessment. The statistical analysis has quality, but the work failed to present the results.

Validity of the findings

The conclusions are flawed. About 30% of what is written in the decision belongs to this topic. I made a note below about this item.

Additional comments

Below, I detail the revisions that I consider important
I suggest compressing the five analyzed elements into a single word. For example, "in five elements"
Line 30 – include which statistical analysis was performed.
Line 41-43 – what is written is still results. Conclude your studies by answering the perspectives of this study. What are the other steps after this study?

Introduction
Line 53-55 – Include a quote in this sentence.
Line 56 - Environmental exposure to heavy metals (...). Include what heavy metals are associated with ASD?
Line 58 - which impacts the normal body physiology (...). What are the types of impacts on the body?
Line 70: Include a quote in this sentence.
Line 71: Compare these results with the standard limits allowed by international or Malaysian agencies.

Material and methods
Line 91 – I recommend dividing this topic into subtopics to attract reading. I recommend initially characterizing the location where the study was carried out. I want to recommend the creation of a summary figure to help the reader know how the stages of your work occurred. The work does not present any images. I recommend creating a map that also indicates the location where the research took place.
Line 118 – include the model and country of production of the equipment.
Line 134-135 – Was any standard reagent used to calibrate the equipment? Was there validation of the results? How did it happen?
Line 15-152 – I recommend that it be in the first paragraph of the material and methods item.

Results
The results are great, but everyone is focused on tables. The authors write practically all the information in the tables, but they should be more succinct and only mention the main results in the text and then (table x, y or z).
The table titles are vague and need to be self-explanatory. Please rephrase them all.
I recommend transforming Table 4 into a figure. It is better for understanding and appealing to the reader.
In all tables, what is the information in parentheses? It's not self-explanatory.

Line 155-156 – I recommend tackling one table at a time rather than all simultaneously. Address what is most attractive in each table.
Line 157-158- A total of 155 children (81 with ASD and 74 children that 158 did not have autism) participated in the study. This sentence should be in material and methods.
Lines 158-159: Does the sentence refer to Table 1??? I believe so, so it has to be included (Table 1).
Line 160-187 – Where do these results come from? From which table? Reread and always include at the end of the sentences which table you are referring to (Table 1) (Table 2) (Table 3).

Table 1
Avoid acronyms within the table. Or, place an explanatory text below the table.
About "age groups". Why did you only create these two ranges? Could have included more categories. A 31-year-old is very different from a 70-year-old.
Income Classification: B40, M40 and T20. What do these terms mean?
Outside Kuala Lumpur. Can you specify which location?
Children's background
age group
More than four years old – up to what age?? In my country, a child is up to 12 years old. And a 5-year-old is different in many ways from a 12-year-old.
Advanced Maternal Age
35 years old and below
More than 35 years old
Over 35 years until when? Is it possible to subdivide into other smaller groups?
PICA – What is it?

Table 2. I recommend removing the standard deviation or error inside the parentheses. And also standardize the tenth place.
Table 3. I needed help understanding the table. And I wonder why the authors repeat values mentioned in previous tables.

Discussion
When making Table 4 in figure format, I suggest you include the maximum level allowed by the country's regulatory agency instead of having this information only in the discussion.
I didn't see a discussion about table 1 and table 2.
I recommend discussing it with data from the literature, as this has often been done, but closing it immediately within the discussion.
This topic is still fragile for the creation of a publication.

Conclusion
Lines 331-341 are the results.
Lines 342-345 are the conclusion. Also, indicate the future perspective and next steps.

References
Standardize formatting.

---

## Round 0.2 · Major Revisions

Dear authors,

At this stage your work requires substantial revisions.

You should go back to the "start", think thoroughly and point-by-point about the reviewers' input. This, in order to achieve sufficient scientific soundness and transparency. The work is important for the scientific community. But still needs essential corrections. Clarify & complete your methods, results and improve discussion!

Reviewer 1 ·

Basic reporting

The paper is generally well-written

Experimental design

The design is generally fine

Validity of the findings

Abstract: explain why you state As, Pb, and Cd values are lower than certain values – what is important about those levels? Same comment for the statement in the results section.

In the methods section, clearly state the inclusion criteria (age, ability to talk for the control group, autism diagnosis for the autism group). How was the autism diagnosis made, and was there any verification of it?
Please clarify the geographical location of the participants – it appears that the ASD group was all attending one school? Could that affect their level of exposure to toxic metals?
Table 1 –The title in the table says no significant differences, but the text reports several statistical differences. Correct the table and list the p-values in the table for each significant difference.

There are major differences between the ASD group and the control group in terms of geographical location and in terms of education. Please calculate the levels of each toxic metal for the subgroups of education level and the subgroups of geographical location (inside and outside Kuala Lampur). I wonder if the difference in toxic metal levels is primarily due to differences in education level and/or differences in geographical location. A statistical analysis should also be done to control for those factors, to determine if ASD diagnosis, education level, and/or geographical location are the primary factors affecting level of toxic metals.

Overall there are many other studies that the authors have not referenced which report higher levels of lead in children with ASD. A search of pubmed for “autism lead urine” revealed 51 hits - See for example this meta-analysis: https://pubmed.ncbi.nlm.nih.gov/37755763/

The authors need to do a more thorough literature review of other studies of toxic metals in ASD

---

## Round 0.3 · accepted · Accept

Dear authors, I am happy to let you know that your paper has been accepted for publication in PeerJ.

Reviewer 1 ·

Basic reporting

The writing is clear

Experimental design

The case-control design is fine.

Validity of the findings

the findings appear to be fine.

Additional comments

The authors have addressed my comments and the paper is now ready for publication.

Reviewer 2 ·

Basic reporting

Dear editor

The manuscript has been revised and I believe it is ready to be published.

Congratulations to the authors.

Experimental design

Dear editor

The manuscript has been revised and I believe it is ready to be published.

Congratulations to the authors.

Validity of the findings

Dear editor

The manuscript has been revised and I believe it is ready to be published.

Congratulations to the authors.

Additional comments

Dear editor

The manuscript has been revised and I believe it is ready to be published.

Congratulations to the authors.